# Silver Nanoparticles Densely Grafted with Nitroxides as a Recyclable Green Catalyst in the Selective Oxidation of Alcohols

**DOI:** 10.3390/nano12152542

**Published:** 2022-07-24

**Authors:** Agnieszka Krogul-Sobczak, Natalia Pisarek, Piotr Cieciórski, Elżbieta Megiel

**Affiliations:** Faculty of Chemistry, University of Warsaw, Pasteura 1, 02-093 Warsaw, Poland; akrogul@chem.uw.edu.pl (A.K.-S.); nj.pisarek@student.uw.edu.pl (N.P.); p.cieciorski@uw.edu.pl (P.C.)

**Keywords:** surface modification, silver nanoparticles, nitroxides, catalysis

## Abstract

The selective oxidation of alcohols, leading to appropriate aldehydes, is widely recognised as one of the most important reactions in organic synthesis. With ever-increasing environmental concerns, much attention has been directed toward developing catalytic protocols that use molecular oxygen as an oxidant. An ideal green oxidation process should employ a highly active, selective and recyclable catalyst that can work with oxygen under mild conditions. This paper presents a successful application of densely grafted silver nanostructures with stable nitroxide radicals (N-AgNPs) as an effective, easily-recovered and regenerable catalyst for the selective oxidation of alcohols. The fabricated ultra-small and narrow dispersive silver nanoparticles have been fully characterised using physicochemical methods (TEM, DLS, XPS, TGA). N-AgNPs have been successfully applied to oxidise several model alcohols: benzyl alcohol, 4-pyridinemethanol, furfuryl alcohol, 1-phenyl ethanol, n-heptanol and allyl alcohol under mild conditions using oxygen as a stoichiometric oxidant. Notably, the fabricated nitroxide grafted silver nanoparticles (N-AgNPs) were reused more than ten times in the oxidation of a series of primary alcohols to corresponding aldehydes under mild conditions with very high yields and a selectivity close to 100%.

## 1. Introduction

The selective oxidation of alcohols is an essential reaction in the synthesis of organic compounds. These compounds are ubiquitous intermediates in the synthesis of agrochemicals, pharmaceuticals and fine chemicals. Modern organic chemistry textbooks present various methods of oxidising alcohols using chromium [1] and manganese oxides [2], pyridinium chlorochromate [1], Jones reagent [3], “activated DMSO” [4,5] and hypervalent iodine reagents [6,7]. However, these methods cannot be considered environmentally benign as toxic and corrosive oxidising agents are employed. Additionally, applying these methods in pharmaceutical production poses severe problems in accomplishing restrictive purity requirements.

In recent years, the demand for methods suitable for large-scale, incredibly environmentally benign applications has increased, principally within the pharmaceutical industry. Extensive efforts toward catalytic systems for selective alcohol oxidation using molecular oxygen, an ideal stoichiometric oxidant, are made. Searching for a system specially inscribed as green is still one of the biggest challenges in modern chemistry.

N-oxyl compounds, also named nitroxides, can be successfully applied as selective catalysts to oxidise alcohols using oxygen as a stoichiometric oxidant [8,9,10,11,12]. This group of stable radicals includes many aliphatic and cyclic organic compounds with >N-O• moiety, which may be reversibly oxidised to oxoammonium ion in a one-electron process.

It is noteworthy that the oxoammonium salts are relatively strong two-electron oxidising agents and that they can be used in many oxidation reactions in organic synthesis [13,14].

The most prominent representative among cyclic nitroxides is 2,2,6,6-tetramethylpiperidine-N-oxyl (TEMPO). Sheldon’s group [15,16] and later Stahl’s [17,18] developed elegant catalytic systems composed of copper(I) salt, chelating nitrogen ligands such as a bipyridyl (bpy) and TEMPO ((bpy)/Cu^I^/TEMPO) for selective oxidation of alcohols to aldehydes with very high yield under mild conditions. It was shown that Sheldon’s catalytic system works very effectively for the oxidation of primary benzylic and allylic alcohols, but limitations have been observed in the case of aliphatic alcohols [16]. In the catalytic system proposed by Stahl and co-workers, N-methylimidazole (NMI) as a catalytic base was employed (instead of t-BuOK in the Sheldon system) and acetonitrile (MeCN) as a solvent [17]. This catalytic system gave excellent selectivity in the aerobic oxidation of a broad range of primary alcohols such as benzylic, allylic and aliphatic with various functional groups.

Generally, in the Stahl oxidation, by using a (bpy)/Cu^I^/TEMPO/NMI system, two main half-reactions may be highlighted: (i) oxidation of Cu(I) salt and reduced N-oxyl (TEMPOH) and (ii) alcohol oxidation by Cu(II) chelated by bpy and NMI and TEMPO. Interestingly, the mechanistic studies performed by Stahl and co-workers showed that oxoammonium cation TEMPO^+^ as an intermediate is not involved in alcohol oxidation, but Cu(II) chelated with bpy and NMI together with TEMPO act collectively as a two-electron oxidising agent [18]. High oxidation yields were obtained when 5 mol% of TEMPO, 5 mol% of bpy, 5 mol% of Cu(OTf) and 10 mol% of NMI were applied.

The immobilisation of TEMPO onto inorganic surfaces for obtaining recyclable catalysts was studied intensively during the last three decades. A vast number of materials were tested for selective oxidation of alcohols [19]. Silica grafted with TEMPO was employed as a heterogeneous recyclable catalyst in the oxidation of alcohols [20,21,22,23,24]. However, due to the low grafting density typically obtained for such materials and their poor stability, efficiency and recyclability in alkaline solutions, catalysts of this type are usually insufficient. Significantly better results were obtained in the case of silica surfaces prepared using sol-gel techniques [25,26,27,28,29,30]. Fullerenes [31,32,33,34], graphene and graphene oxide [35,36] grafted with TEMPO were also applied in the selective oxidation of alcohols. All these mentioned above methods of TEMPO immobilisation enable the reactions to perform in a heterogeneous system. In contrast, the attachment of an organocatalyst on nanoparticles enables the obtention of a colloidal system, and a semi-homogeneous process may occur, especially when small/ultra-small particles are used as the carriers. Thus, the immobilisation of catalysts onto nanoparticles gives a significantly better performance [37].

Among the nanoparticles, metal nanostructures have attracted the most attention as remarkable catalysts due to their high activity and stability as well as tunable functionalisation [38]. Recently, we reported the successful application of gold nanoparticles as a carrier for TEMPO radical in the electrocatalytic oxidation of benzyl alcohol [39]. We also developed several efficient procedures to prepare ultra-small, stable silver nanoparticles (AgNPs) densely covered with nitroxyl radicals [40,41,42].

As shown, AgNPs can also catalyse the oxidation of alcohols by oxygen adsorption on silver surfaces [43]. Thus, we expected that using AgNPs grafted with TEMPO as an oxidation catalyst should result in a synergistic effect between these two components.

Herein, we report the successful application of silver nanoparticles densely grafted with nitroxide radicals (N-AgNPs) for highly selective oxidation of a wide range of primary alcohols. The most important benefits of applying the silver nanoparticles are the following: (i) recovery and reuse of an organocatalyst, (ii) lowering the required amount of organocatalyst—a synergistic effect between AgNPs and TEMPO is likely to occur and (iii) higher purity of the obtained products—TEMPO might be easily removed from the post-reaction mixture.

## 2. Materials and Methods

### 2.1. Materials

Sodium borohydride (NaBH_4_), silver(I) nitrate (AgNO_3_), sodium chloride (NaCl), 2,2,6,6-tetramethylpiperidine-1-oxyl (TEMPO), 4-hydroxy-TEMPO (TEMPOL), benzyl alcohol, 4-pyridinemethanol, furfuryl alcohol, 1-phenylethanol, n-heptanol, tetrakisacetonitrile copper(I) triflate ([Cu(MeCN)_4_](CF_3_SO_3_)), 2,2‘-bipyridyl (bpy), 1-methylimidazole (NMI) and all solvents were purchased from Sigma-Aldrich (purity ≥ 97%), and used as received (supplied by Sigma-Aldrich (Steinheim, Germany)). Bis[2-(4-oxy-2,2,6,6-tetramethylpiperidine-1-oxyl)ethyl] disulfide (DISS) was prepared according the procedure described in our earlier papers [39,42]. Argon and molecular oxygen were purchased from Air Products (purity ≥ 99,999%, Poland) and used as received. Milli-Q ultrapure water (resistivity 18.2 MΩ∙cm^−1^) was used throughout the experiments.

### 2.2. Techniques

The UV–vis absorption spectra of N-AgNPs were recorded using a Cary 50 UV/Vis spectrophotometer in acetone in the 200–800 nm range with a 1 cm quartz cell. The concentration of the solutions was between 0.5 and 1.5 mg∙ml^−1^. The thermogravimetric (TG) measurements were performed under N_2_ using Q50-1261 TA Instruments (New Castle, DE, USA) with temperature compensated thermobalance (precision ±0.01%), heating rate = 5 K∙min^−1^. Weight loss during thermal decomposition of N-AgNPs was determined in the temperature range of 20–800 °C. TG measurements were performed in a platinum pan, and the weight of the sample was around 2 mg. The transmission electron microscopy (TEM) observations have been carried out using JEM 1400 JEOL Co. microscope at 120 kV acceleration voltage. The samples were obtained by casting the acetone solution of materials onto a carbon-coated copper microgrid (200 mesh) and air-dried overnight. The electron spin resonance (ESR) spectroscopy experiments were performed at room temperature in acetone solutions with an X-band (9.7 GHz) using a microESR (Bruker) spectrometer. The X-ray photoelectron spectroscopy (XPS) analysis was performed using PHI 5000 VersaProbe (Scanning ESCA Microprobe ULVAC-PHI), equipped with an Al Kα source (1486.6 eV, power 23 W). CasaXPS (version 2.3.19) software was used to deconvolution XPS signals. XPS data were calibrated using the binding energy of C 1s = 284.6 eV (C-C bond) as the internal standard. Analysis of the post-reaction mixture was performed using GC-FID Agilent 7820A with HP-5 column (length = 30 m, diam. = 0.32 mm, film = 0.25 µm), with the following parameters: inlet temperature = 350 °C, detector temperature = 300 °C, oven temperature = 80–180 °C (see Appendix A). Yield, selectivity and TON values were determined based on the results of GC-FID analyses.

### 2.3. Preparation of Nitroxide-Coated Silver Nanoparticles (N-AgNPs)

N-AgNPs on a scale of hundreds of milligrams were synthesised using a lab set composed of the jacketed reactor (with a capacity of 0.5 L), mechanical stirrer and thermostat. The optimisation of the process of AgNPs synthesis using this lab set is described in our earlier papers [40,42].

Briefly, 576.8 mg of DISS (1.124 mmol) was dissolved in 300 mL of DMF in the reactor, and the mixture was bubbled with argon (for 20 min) and stirred using a mechanical stirrer (1600 RPM); the temperature of the cooling medium was set at −20 °C. When the mixture achieved the set temperature, 0.949 mL of AgNO_3_ solution with concentration 1 M (0.949 mmol) was injected and stirred for an additional 5 min. After this time, a solution of NaBH_4_ (64 mg, 1.702 mmol) in DMF (120 mL) was added using a peristaltic pump (Thermo Fisher Scientific) with a pumping velocity of 1.7 mL∙min^−1^. During the addition of the solution of NaBH_4_, the stirring of the mixture was accelerated to 3000 RPM. After adding the reducing agent solution, the mixture turned from light orange through pale-yellow to a dark brown colour; the stirring was continued with a velocity of 1600 RPM for the next 2 h maintaining temperature in the range of −18 °C to −20 °C. Afterwards, the post-reaction mixture was mixed with ultra-pure water (500 mL). A pinch of NaCl was added to precipitate the fabricated nanoparticles. The obtained suspension was sonicated (2 min), centrifuged (10.000 RPM, 10 min), the supernatant was discarded and the precipitate was washed profusely several times with water (ultra-pure). The obtained solid was dried in a vacuum oven at 50 °C for 24 h (20 mbar). Thin-layer chromatography (TLC) confirmed the absence of impurities and not-attached ligands in the solution of the prepared N-AgNPs. Finally, the grey, solid product was obtained (118 mg, yield = 78%, calculated based on silver content from TG analysis).

### 2.4. Representative Procedure for the Oxidation of Alcohols by (bpy)Cu^I^/N-AgNPs Catalytic System

The oxidation of the alcohols was conducted according to the procedure described by Stahl et al. [17] with some modifications and a lower molar ratio of organocatalyst to reactants.

In a 20 mm culture tube, 1 mmol of the alcohol was combined with anhydrous MeCN (0.5 mL) and the following solutions were added: [Cu(MeCN)_4_](CF_3_SO_3_) (0.03 mmol in 0.05 mL MeCN); 2,2’-bipyridyl (0.03 mmol in 0.5 mL MeCN); N-AgNPs (4.5 or 9 mg, i.e., amount containing, respectively, 0.0042 or 0.0084 mmol of TEMPO, in 0.5 mL MeCN); N-methylimidazole (0.06 mmol in 0.5 mL MeCN). The reaction mixture was magnetically stirred open to the air or fitted with a septum and air balloon (or O_2_). The reactions were performed at room temperature for 2.5–6 h, depending on the experiment. Upon completion of the reaction, 250 µL of MeCN was added to 250 µL of the reaction mixture. The obtained mixture was centrifuged (6000 RPM, 1 min), and then a sample (1 μL) of supernatant fluid was taken and analysed by GC-FID. Nanoparticles were transferred back to the reaction mixture, and the whole post-reaction mixture was centrifuged (6000 RPM, 6 min). After removing the supernatant, nanoparticles were suspended using sonication in 0.5 mL of MeCN for the next cycle.

## 3. Results and Discussion

### 3.1. Synthesis and Characterisation of N-AgNPs

In this study, silver nanoparticles grafted with stable nitroxide radicals (N-AgNPs) were synthesised as a platform for the oxidation of alcohols. The stabilising ligand and nitroxide radicals source was bisnitroxide disulfide containing TEMPO moieties (DiSS). N-AgNPs were synthesised using a one-pot and one-phase procedure with AgNO_3_ as a silver ions precursor, DiSS as a ligand, NaBH_4_ as a reducing agent and DMF as a solvent (Figure 1).

The UV-vis spectrum for N-AgNPs in dimethylformamide (DMF) is presented in Figure 1a. The Surface Plasmon Resonance (SPR) band observed in the spectrum is symmetrical and narrow (with the maximum at 415 nm), indicating that metallic cores of nanoparticles are spherical and small with narrow size distribution. These conclusions are consistent with TEM analyses results. Figure 2b shows the selected TEM micrograph of the synthesised N-AgNPs, and the inset in Figure 2b shows a histogram presenting their size distribution prepared on the base of this image. The average diameter of the nanoparticles is d = 5.3 ± 1.6 nm.

DLS analysis allowed us to investigate the size and dispersity of synthesised N-AgNPs, including a stabilisation layer attached to the surface of the nanoparticles. The size distribution calculated by the number is shown in Figure 1c. The average size is 22.1 ± 3.96 nm; thus, the size dispersity is narrow. The thickness of the stabilisation layer with the solvation layer can be calculated as ca. 15 nm (from TEM and DLS measurements).

The ESR spectrum of obtained nanoparticles is shown in Appendix A (for comparison, the ESR spectrum of ligand (DiSS) is also presented). The ESR measurements confirm the presence of nitroxyl radicals on the surface of N-AgNPs. The pattern of the spectra of N-AgNPs originates from the strong interactions between radicals closely packed on the metal surface [39,42].

X-ray photoelectron spectroscopy (XPS) was used to characterise the composition and structure of fabricated nanoparticles. The quantitative data from the XPS analysis are summarised in Table 1 (survey XPS spectrum is shown in Appendix A).

Figure 2a displays the XPS spectrum of Ag 3d. The spectrum consists of two intense peaks with binding energies 373.8 and 367.8 eV and two peaks with significant lower intensity at 374.6 and 368.6 eV, all with the same full width at half maximum (FWHM) in each pair (1.03 and 1.84 eV). These two doublets correspond to 3d_5/2_ and 3d_3/2_ orbitals in the different atomic surroundings. A distinct characteristic for silver nanoparticles spin-orbit separation of 3d orbitals equal to 6 eV [44] is observed between the signals in both pairs. The peaks with smaller intensity are shifted to higher energies than observed for a bulk silver Ag^0^ (368.2 eV for 3d_5/2_ and 374.2 eV for 3d_3/2_). Such bond energies of Ag 3d orbitals were observed for thiolate bonds with a silver (Ag-S) [42,45]. These silver atoms, which form the Ag-S bond, consist of ca. 20% of all silver atoms in the sample. It shows that the fabricated nanoparticles are densely grafted with thiolate-connected ligands.

The S 2p spectrum of the synthesised nanoparticles is shown in Figure 2b. One broad peak can be deconvoluted into four contributions at 164.0, 162.8, 162.3 and 161.1 eV with the same FWHM equalling 1.6 eV. These peaks correspond very well to the BDE of orbitals 2p_3/2_ and 2p_1/2_ in two distinct sulphur species adsorbed on the silver surface of the nanoparticles. The fitted peaks at 161.1 and 162.3 eV come from 2p_3/2_ and 2p_1/2_ orbitals of sulphur bound as a thiolate bond [45]. The second doublet at higher energies (162.8 and 164.0 eV) indicates the presence of sulphur connected with the silver surface via disulphide bonds [44]. The ratio of sulphur connected via thiolate bonds to disulphide ones is 7:1, which corresponds to ca. 6% of the last type of bonds on the silver surface.

The XPS spectrum of N 1s can be deconvoluted into three peaks: 398.6, 400.1 and 402.0 eV, with the same FWMH, equalling 1.45 eV. The first components correspond very well to these observed for free nitroxide moieties [42,46]. While the presence of the third peak at 402 eV indicates that part of the nitroxide groups is involved in forming bonds with the silver. The possibility of creating such bonds of nitroxides with gold and silver was postulated in our earlier papers [39,42,47]. The quantitative data from XPS indicate that 25% of nitroxide groups is involved in Ag-ON bonds. Thus, this part of these groups is most probably non-active in alcohol oxidation. C 1s and O 1s XPS spectra are shown in the Appendix A.

Thermogravimetric analysis (TG) permits the study of the material’s thermal stability and, in the case of metal nanoparticles, determines organic fraction content. The thermograms obtained for N-AgNPs and ligand used for their preparation are presented in Appendix A. From TGA results and assumptions on the shape of nanoparticles, the density of grafting ligands on nanoparticles could be calculated as 10 per nm^2^. Such a high density of nitroxide molecules on nanoparticles is crucial for effective catalysis with their participation.

TGA and XPS allowed us to determine the content of total and catalytically available TEMPO radicals in the fabricated nanomaterial. Considering that 75% of total TEMPO radicals is readily available for catalysis (see above discussion on the XPS spectrum N 1s), the concentration is 0.93 µmol of TEMPO radicals per 1 mg of N-AgNPs.

### 3.2. Catalytic Oxidation of Alcohols

In our further experiments, we investigated the application of N-AgNPs in the oxidation of alcohols as a recyclable catalyst. It was used in the consecutive catalytic cycles in the oxidation of the following: benzyl alcohol (a model primary aromatic alcohol); 4-pyridinemethanol and furfuryl alcohol (primary aromatic heterocyclic alcohols); 1-phenylethanol (a model secondary aromatic alcohol); n-heptanol (a model primary aliphatic alcohol) and allylic alcohol (model alcohol with a double bond in its structure) (Figure 3).

#### 3.2.1. Catalytic Activity of N-AgNPs in the Consecutive Cycles of Oxidation of Benzyl Alcohol with the Various Source of Oxygen

Catalytic tests were started by examining the influence of several sources of oxygen on the rate of the process (expressed as the yield of the main product) and with various quantities of N-AgNPs. First, we chose the amount of N-AgNPs containing the content of catalytically active TEMPO radicals exhibiting satisfactory effectiveness. For this reason, 4.5 mg of N-AgNPs (containing 0.0042 mmol of TEMPO and 3.03 mg of silver) was used, and the results were compared with the results obtained for twice the amount of N-AgNPs. The results are presented in Table 2.

Considering the results presented in Table 2, we observe very high yields in the first catalytic cycle regardless of the source of the oxidising agent (entries 1 and 6 in Table 2). The obtained yields are the same as in the case of employment of free TEMPO radical with the same amount as immobilised onto nanoparticles (see Appendix A). Thus, the designed way of nitroxide immobilisation did not deteriorate its activity in the catalytic system. Moreover, we observe a positive influence of O_2_ (from a balloon) instead of air on the reaction yield in the first cycle. However, a significant yield decrease is observed in the second cycle if the reaction is performed in pure oxygen or air from a balloon. A continuous and significant reduction in yield in the presence of pure oxygen indicates that an excess of oxygen irreversibly oxidises N-AgNPs. These facts can be explained by the oxidation of sulphur in the nanocatalyst leading to the break of the Ag-S bond between nitroxide ligand and silver surface. Therefore, in the consecutive cycles, density of nitroxides grafted onto the nanoparticles decreases and, consequently, the yield of the selective oxidation of alcohol also decreases. In the case of other sources of oxygen in which its content is significantly lower, the sulphur oxidation rate is lower, and the decrease in yield is also considerably lower. It was reported that thiol-capped nanoparticles are stable in air for a long time, but under an oxygen atmosphere, ligands are released due to sulphur oxidation [48].

A few surprising results observed for yields of reactions performed in the presence of air from a balloon (namely, significantly higher yield noticed in the 5th cycle than those obtained in cycles 2–4) suggest that irreversible oxidation of N-AgNPs is not the main reason for the decreasing yield in the 2nd–4th cycles. Otherwise, we would also observe a lower yield in the 5th one, but the opposite effect—a higher yield—is noticed in this cycle. Increasing yield is also observed for the 7th and 9th cycles. It must be emphasised that before these cycles, N-AgNPs were kept in acetonitrile for a longer time (two days) compared to other cycles (where N-AgNPs were kept in acetonitrile for one night). Perhaps, after a more extended period, more nanoparticles might be suspended in acetonitrile (from those attached to the wall of the reaction tube). Therefore, we extended the sonication time to ensure that the whole nanomaterial was suspended in further experiments.

Considering the reaction performed in the open vial (under atmospheric air used as a source of oxygen), we do not observe a drastic yield decrease in the second cycle. Moreover, the results presented in Table 2 indicate that N-AgNPs are active for many cycles in the oxidation of benzyl alcohol under atmospheric air. Indeed, the yield of aldehyde gradually decreases, probably due to the ligand and/or N-AgNPs loss during purification and recyclability of the catalyst. However, the catalyst exhibits a relatively high activity up to nine cycles (yield above 50%, 80% during the first three cycles). Thus, we chose this source of oxygen as the most appropriate to scope other alcohols. This solution is also the most environmentally benign.

#### 3.2.2. Catalytic Activity of N-AgNPs in the Consecutive Cycles of Oxidation of Selected Alcohols

After satisfactory results for benzyl alcohol, we determined the reusability of N-AgNPs in catalytic oxidation of other model alcohols, namely, 4-pyridinemethanol, furfuryl alcohol, 1-phenylethanol, n-heptanol and allylic alcohol. In addition, we tested various reaction conditions such as time of reaction and amount of N-AgNPs. The selected results of our optimisation experiments are presented in Table 3.

Firstly, we showed that, in consecutive catalytic cycles, it is still possible to obtain a very high yield; enough time extension is only required. For example, oxidation of benzyl alcohol processing for 4 h allows the reception of 100% conversion and 96% yield in the second catalytic cycle (Table 3—experiment 1 and Appendix A). We continued these experiments (performing consecutive cycles for 4 h) until the yield was below 80%. As it turned out, the yield was still high after the 6th catalytic cycle and equalled 78% (Figure 4). We used the reaction conditions of benzyl alcohol oxidation for furfuryl alcohol (experiment 4). For the latter, a yield below 80% was noted in the second cycle; however, it was still very high, higher than 70% (Table 3, entry 15). A similar trend (yield below 80% in the second cycle but still very high, higher than 70%) was observed for oxidation of benzyl alcohol and 4-pyridnemethanol in the presence of a lower amount of N-AgNPs (Table 3, entries 9 and 12—experiment 2 and 3, respectively). Although high conversions and yields are obtained for primary aromatic alcohols, we noted significantly less satisfactory results for n-heptanol, 1-phenylethanol and allylic alcohol (Table 3). Moreover, substantially lower selectivity (below 90%) of the catalyst in the oxidation of n-heptanol and 1-phenylethanol were observed. Such results indicate that N-AgNPs are selective towards primary aromatic alcohols. The differences between results obtained for primary and secondary alcohols may be a consequence of the deactivation of N-AgNPs by the latter. To confirm or reject this hypothesis, we performed oxidation of furfuryl alcohol—additional experiments (Table 3—experiment 8, see footnote g under Table 3)—in the presence of N-AgNPs recycled from the reaction mixture after the first cycle of oxidation of 1-phenylethanol. Surprisingly, a high yield of furfural was observed for a few cycles. Such results indicate that N-AgNPs are still catalytically active (but neither towards secondary aromatic alcohols nor aliphatic and allylic ones). Based on the obtained results, we can conclude that the designed catalytic system based on the N-AgNPs is highly selective and active toward primary aromatic alcohols.

In our optimising experiments, two amounts of nanocatalyst were applied: namely, 4.5 mg or 9 mg containing 0.0042 mmol or 0.0084 mmol of nitroxide radicals, respectively. Both are significantly lesser than those used in the catalytic system developed by Stahl and co-workers [17] and gave high yields *(*Table 3). However, better outcomes were obtained for a higher amount of nanocatalyst in several consecutive cycles. Thus, we decided to use the amount of nanocatalyst in further experiments containing 0.0084 mmol of nitroxide radicals. Among the applied reaction time, 2.5 h seems to be optimal. Under these conditions, we compared the catalytic activity of N-AgNPs in the consecutive cycles of oxidation of the most reactive alcohols; namely, benzyl alcohol, 4-pyridinemethanol and furfuryl alcohol (Table 4 and Figure 5). Due to the presence of heteroatoms in the molecules of 4-pyridinemethanol and furfuryl alcohol (nitrogen and oxygen, respectively) the formation of bonds with silver atoms is enabled.

We expected the stabilisation of AgNPs in consecutive catalytic cycles of their oxidation. Although we do not observe significantly higher yields for these alcohols than for benzyl alcohol in the first cycles, the effect of stabilisation is visible in further cycles. The oxidation yield of 4-pyridinemethanol with N-AgNPs is higher or around 50% in 15 consecutive catalytic cycles (see Table 4 and Appendix A). Such results indicate that 4-pyridinemethanol stabilises AgNPs, which plays a crucial role after a few cycles.

Compared with literature data for Cu/TEMPO catalytic systems in the oxidation of benzyl alcohol, the obtained yields in our experiments are similar or better, and TON values are significantly higher. Importantly, the amount of employed organocatalyst (TEMPO) is even 20 times less than that used in earlier catalytic systems (Appendix A).

TEM studies confirmed the stability of the developed nanocatalyst in several cycles in the oxidation of benzyl alcohol. Figure 6 displays TEM micrographs of nanoparticles separated from the post-reaction mixture after the second, fourth and eleventh catalytic cycle. As it can be seen, after the second cycle sizes and morphology of nanoparticles are not changed; after the fourth cycle, the fraction of bigger nanostructures is visible; after the eleventh cycle, aggregated structures with a size of even 100 nm are visible but still small nanoparticles are present and thanks to this the catalyst is still active.

Apart from high yields and conversions, we observe high selectivities of our catalytic system in the oxidation of primary aromatic alcohols (100% towards furfural and above 90% for benzaldehyde and 4-pyridinemethanal). Interestingly, 100% selectivity was noted for both benzaldehyde and 4-pyridinecarboxaldehyde when reactions were performed for a longer time (Table 3, entries 7–12). Stahl and co-workers observed earlier high selectivities (>90%) in oxidation alcohols using (bpy)Cu^I^/TEMPO/NMI system. However, TON values obtained for our catalytic system are significantly higher than reported in the literature. For example, we got TON = 114 for the oxidation of benzyl alcohol (calculated on the amount of TEMPO radicals), whereas for Stahl’s system [17], the TON is only 19. Meanwhile, Koskinen’s catalyst system [49] made it possible to achieve good yields in the oxidation of alcohols with TEMPO with TON equals 26. Thus, the proposed catalytic system obtains excellent outcomes using significantly lower amounts of nitroxide catalyst.

Notably, the obtained yields are the same as in the employment of free TEMPO radical with the same amount as immobilised onto nanoparticles (see Table 3 and Appendix A). Thus, it shows that the designed way of nitroxide immobilisation did not deteriorate its activity in the catalytic system. It may also indicate the synergistic effect between AgNPs and TEMPO.

Thanks to the designed catalyst system, the oxidation of the primary alcohols can be performed effectively using oxygen from the air as an oxidising agent. It gives an outstanding atom efficiency of 76% for allyl alcohol and 85% for benzyl alcohol. As the organocatalyst in the designed system is grafted onto AgNPs, it can be easily separated from the post-reaction mixture (via centrifugation), and the final product can be easily purified from the organocatalyst.

## 4. Conclusions

Oxidation of alcohols with air (as a source of oxidising agent) is of great interest nowadays. Notably, the development of a more effective, selective and regenerable catalyst for selective oxidation of alcohols is a big challenge in modern chemistry.

During this work, we developed and synthesised (using a one-pot procedure) an effective catalyst for selective oxidation of primary aromatic alcohols. The catalyst is based on a hybrid nanomaterial (N-AgNPs) consisting of ultra-small silver nanoparticles densely grafted with stable nitroxide radicals (10 nitroxide moieties per 1 nm^2^). The obtained nanomaterial was fully characterised by a series of analytical methods (UV-Vis, TEM, DLS, ESR, XPS and TG). The symmetrical and narrow Surface Plasmon Resonance (SPR) band observed in the UV-Vis spectrum (with the maximum at 415 nm) indicates that metallic cores of nanoparticles are spherical and small with a narrow size distribution. The small diameter of the nanoparticles with a narrow size distribution is confirmed by TEM (d = 5.3 ± 1.6 nm) and DLS (22.1 ± 3.96 nm, including stabilisation layer with the solvation layer ca. 15 nm) measurements. The ESR measurements confirm the presence of nitroxyl radicals on the surface of N-AgNPs and indicate the strong interactions between radicals. The nanoparticles’ composition and structure were analysed using XPS and complemented by TG analysis. The XPS measurements show that the fabricated nanoparticles are densely grafted with organic (thiolate connected) ligands. From TGA results and assumptions on the shape of nanoparticles, the density of grafting ligands on nanoparticles could be calculated as 10 per nm^2^. Such a high density of nitroxide molecules on nanoparticles is crucial for effective catalysis with their participation. TGA and XPS allowed us to determine that the concentration of TEMPO radicals readily available for catalysis is 0.93 µmol of TEMPO radicals per 1 mg of N-AgNPs.

The fabricated N-AgNPs were employed as effective and recyclable catalysts in the oxidation of selected primary aromatic alcohols, benzyl alcohol, 4-pyridinemethanol and furfuryl alcohol. The results obtained in a series of optimisation tests for benzyl alcohol indicate that the air is the most appropriate oxidising agent source. We showed that it is possible to obtain very high yields in consecutive catalytic cycles; enough time extension is only required (e.g., 4 h for benzyl alcohol makes it possible to receive a yield of around 80% for six catalytic cycles). The high effectiveness of our catalyst was also observed during the oxidation of benzyl alcohol, 4-pyridinemethanol and furfuryl alcohol for a reaction time of 2.5 h. It should be emphasised that in the oxidation of 4-pyridinemethanol, N-AgNPs were highly active for more than 15 consecutive cycles. However, less satisfactory results were noted for n-heptanol, 1-phenylethanol and allylic alcohol. Based on the obtained results, we concluded that the designed catalytic system based on the N-AgNPs is highly active toward primary aromatic alcohols. Apart from high yields and conversions, we observe in the studied reactions very high selectivities (close to 100%).

N-AgNPs, as a catalyst in the selective oxidation of alcohols, exhibit much higher activity (expressed as TON) than many other systems described in the literature. It makes it possible to decrease the amount of organocatalyst in the reaction medium. At the same time, grafting the catalyst onto nanoparticles enables the use of it repeatedly, and the obtained products can be easily purified.

In summary, the proposed catalytic system based on the N-AgNPs gives a possibility to perform selective oxidation of primary aromatic alcohols towards aldehydes in a very effective and, at the same time, environmentally friendly manner.

## Data Availability

The raw/processed data are available upon reasonable request.

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
