# Peer review of "Silver Nanoparticles Densely Grafted with Nitroxides as a Recyclable Green Catalyst in the Selective Oxidation of Alcohols"

_nanomaterials, 2022, doi:10.3390/nano12152542_

Round 1

Reviewer 1 Report

The manuscript reports the selective oxidation of some aromatic and aliphatic alcohols to the corresponding aldehyde by using silver nanoparticles grafted with nitroxides as  a recyclable catalyst in MeCN as the solvent. The manuscript can be accepted after major revision, below some suggestions:

The novelty of the manuscript with respect to previous papers (for example the cited references 15-18) must be highlighted. The preparation of the catalyst is not new, the same sample has been previous prepared and characterised by the authors.

The role of Ag must be better clarified. How was the amount of Ag chosen? Samples containing different quantities of Ag should be prepared.

The results reported in Table 2 revealed that by using atmospheric air as a source of oxygen better results are obtained. A tentative explanation should be inserted.

A comparison with literature results obtained with the same substrate in similar experimental condition should be reported.

What is the reaction temperature?

Author Response

Response to Reviewer #1's comments:

Thank you very much for your thorough reading of our work. We are very grateful for all Your comments and suggestions. Our manuscript is significantly better thanks to the changes/corrections made according to Your comments and suggestions.

  1. "The novelty of the manuscript with respect to previous papers (for example the cited references 15-18) must be highlighted. The preparation of the catalyst is not new, the same sample has been previous prepared and characterised by the authors."

To our knowledge, silver nanoparticles have not yet been applied as a support for organocatalyst in the selective oxidation of alcohols. Our current work reports the successful employment of such nanomaterials in these processes. The cited references 15-18 present oxidation of alcohols using the CuI(bpy) system without silver support; meanwhile, references  39-42 show the optimisation of syntheses of silver nanoparticles grafted with nitroxides and application of such materials for the fabrication of bioactive composites. Additionally, the XPS technique used in this work allowed the analysis of nitroxide groups available in the oxidation reactions. It was not discussed in the literature and was not presented in the mentioned our previous papers.

Taking into account the comment of the Reviewer, we decided to add this fragment in the abstract as follows:

This paper presents a successful application of densely grafted silver nanostructures with stable nitroxide radicals (N-AgNPs) as an effective, easily recovered and regenerable catalyst for selective oxidation of alcohols. The fabricated ultra-small and narrow dispersive silver nanoparticles have been fully characterised using physicochemical methods (TEM, DLS, XPS, TGA).

  1. "The role of Ag must be better clarified."

Thank you for the comment; indeed, it is worth introducing some information to clarify silver's role and its application's benefits. The following changes have been introduced to the revised manuscript:

The most important benefits of applying the silver nanoparticles are the following: i) recovery and reuse of organocatalyst, ii) lowering the required amount of organocatalyst – synergistic effect between AgNPs and TEMPO is likely to occur, and iii) higher purity of the obtained products – TEMPO might be easily removed from the post-reaction mixture.         

  1. 2 "How was the amount of Ag chosen? Samples containing different quantities of Ag should be prepared."

Thank you for this comment; the best results in optimizing tests were obtained when we applied 0.76 % of silver (calculated as the mass part of the system), and this amount was used in further experiments. But we also tried smaller quantities of Ag and TEMPO. These experiments are presented in Table 2 (optimizing section) and Table 3.

We have provided an additional statement in the manuscript to explain how the amount of AgNPs was chosen. The following text was introduced in the revised version of the manuscript:

Catalytic tests were started by examining the influence of several sources of oxygen on the rate of the process (expressed as the yield of the main product) and with various quantities of N-AgNPs. First, we chose the amount of N-AgNPs containing the content of catalytically active TEMPO radicals exhibiting satisfactory effectiveness. For this reason, 4.5 mg of N-AgNPs (containing 0.0042 mmol of TEMPO and 3.03 mg of silver) was used, and the results were compared with the results obtained for twice the amount of N-AgNPs.

We have also modified the text in the footnote under Table 3:

Different quantities of N-AgNPs were applied, i.e. 4.5 mg of N-AgNPs containing 0.0042 mmol of nitroxide radicals or 9 mg of N-AgNPs containing 0.0084 mmol of nitroxide radicals (determined by TG and XPS)."

  1. "The results reported in Table 2 revealed that by using atmospheric air as a source of oxygen better results are obtained. A tentative explanation should be inserted."

We are thankful for that important suggestion. We have added the tentative explanation in the revised version of the manuscript, and additional references have been added.

A continuous and significant decrease in yield in the presence of pure oxygen indicates that an excess of oxygen irreversibly oxidises N-AgNPs. These facts can be explained by the oxidation of sulphur in the nanocatalyst leading to the break of the Ag-S bond between nitroxide ligand and silver surface. Therefore in the consecutive cycles density of nitroxides grafted onto the nanoparticles decreases, and consequently, the yield of the selective oxidation of alcohol also decreases. In the case of other sources of oxygen in which its content is significantly lower, the sulphur oxidation rate is lower, and the decrease in yield is also considerably lower. It was reported that thiol-capped nanoparticles are stable in air for a long time, but under an oxygen atmosphere, ligands are released due to sulphur oxidation [48].

[48] Ling, X.; Schaeffer, N.; Roland, S.; Pileni, M.-P. Langmuir 2015, 31, (47), 12873-12882.

  1. "A comparison with literature results obtained with the same substrate in similar experimental condition should be reported."

Thank you for the suggestion. However, we can compare the yield of reactions obtained in the 1st cycle as silver nanoparticles as support for TEMPO have been employed for the first time in our work.

Literature data have been added in Table S6 in Supplementary Information, and a general description has been added to the manuscript as follows:

Compared with literature data for Cu/TEMPO catalytic systems in the oxidation of alcohols, the obtained yields are similar or better, and TON values are significantly higher (Table SI6 in Supporting Information).

Importantly, the amount of employed organocatalyst (TEMPO) is even ca. 20 times less than that used in earlier catalytic systems (Table SI6 in Supporting Information).

  1. "What is the reaction temperature?"

Reactions were performed at room temperature, and such information was included in the footnotes under Tables 2-4 (abbreviation rt).However, indeed it should be given in the procedure description, so we added information in the Materials and Methods section as follows:

The reactions were performed at room temperature.

Reviewer 2 Report

This manuscript summarizes a study on the effectiveness of TEMPO-based catalysts immobilized on Ag nanoparticles for partial oxidation of several types and structures of alcohols.   The study is interesting and well written and presented. 

The results, however, left me with a few questions and comments: 

1) While not familiar with TEMPO before reading this manuscript, I found in literature that it was discovered in ~1960 and it is well known as a catalyst for partial oxidation.   Upon further reading about it, I found that it is not biodegradable and that it may have some toxicity to aquatic life.   While the process is conducted under mild conditions, I wonder if the process meets the criteria for Green Chemistry, which under the US EPA website states:

Green chemistry : 

  • Results in source reduction because it prevents the generation of pollution
  • Reduces the negative impacts of chemical products and processes on human health and the environment
  • Lessens and sometimes eliminates hazard from existing products and processes
  • Designs chemical products and processes to reduce their intrinsic hazards

Questions:  

- Do the authors have evidence of the TEMPO coming off of the silver nanoparticles in the solvent with time of use?   While the yields and TON decreased with cycle number, the TEM pictures show that the nanoparticles grew by a factor of about 20x  (from about 5 nm to more than 100 nm)... and so surface area and availability of active sites may be the reason for lower yields and TON with cycle number.  However, is the TEMPO contained within the larger particles or did it cleave off of the nanoparticles and is contained in the solution? 

- Can the authors explain better the reason or mechanism for nanoparticle growth in acetonitrile at (presumably) room temperature?   Did TEMPO or oxygen catalyze the sintering of the nanoparticles?  Is there anything that can be modified to prevent or reduce the rate of nanoparticle sintering? 

Regarding the conditions of the oxidation reactions:

--> At what temperature was the reaction? I assume it was conducted at room temperature.  

-->  What are the roles of the other constituents in the reaction?  

"....and the following solutions were added: [Cu(MeCN)4](CF3SO3) (0.03 mmol in 161 0.05 mL MeCN); 2,2'-bipyridyl (0.03 mmol in 0.5 mL MeCN); ... N- 163 methylimidazole (0.06 mmol in 0.5 mL MeCN)."

--> Was the reaction rate limited by oxygen diffusion into the solvent? According to Franco and Olmsted (1990), the solubility of oxygen in acetonitrile is 2.42 mM.  Will O2 interact with the other constituents in the reaction mixture? 

-->  Is this oxidation process and catalyst recovery process scalable?  

-->  As stated by the authors:  

"As shown, AgNPs can also catalyse the oxidation of alcohols by oxygen adsorption on silver surfaces [43]. Thus, we expected that using AgNPs grafted with TEMPO as an oxidation catalyst should result in a synergistic effect between these two components."

Did the authors compare the oxidation of the alcohols using the Ag-TEMPO catalysts with just the Ag nanoparticles and with just TEMPO?  What control experiments were conducted to assess the roles of Ag nanoparticles vs TEMPO on the partial oxidation of alcohols? Was any synergism observed? 

Author Response

Response to Reviewer #2's comments:

Thank you very much for your thorough reading of our work. We are very grateful for all Your comments and suggestions. Our manuscript is significantly better thanks to the changes/corrections made according to Your comments and suggestions.

"This manuscript summarizes a study on the effectiveness of TEMPO-based catalysts immobilized on Ag nanoparticles for partial oxidation of several types and structures of alcohols.   The study is interesting and well written and presented. "

Thank you so much for such a positive evaluation of our work.

  1. "The results, however, left me with a few questions and comments: 

1) While not familiar with TEMPO before reading this manuscript, I found in literature that it was discovered in ~1960 and it is well known as a catalyst for partial oxidation.   Upon further reading about it, I found that it is not biodegradable and that it may have some toxicity to aquatic life.   While the process is conducted under mild conditions, I wonder if the process meets the criteria for Green Chemistry, which under the US EPA website states:

Green chemistry : 

Results in source reduction because it prevents the generation of pollution

Reduces the negative impacts of chemical products and processes on human health and the environment

Lessens and sometimes eliminates hazard from existing products and processes"

Designs chemical products and processes to reduce their intrinsic hazards"

Response:

This is a very accurate comment. The main goal of our work was to make one of the most selective and effective catalytic systems for the oxidation of alcohols more environmentally friendly. For this reason, we used ca. 5 to 10 times lower concentration of TEMPO compared to the amount of TEMPO applied in the catalytic system developed by Stahl. Moreover, immobilization of TEMPO significantly eliminates the amount of TEMPO getting into the post-reaction solution. However, leaking of TEMPO from nanoparticles may occur, but the amount of leaking TEMPO is reduced considerably.

In our opinion, the developed catalytic system meets most of the criteria for Green Chemistry.

"Questions:  

- Do the authors have evidence of the TEMPO coming off of the silver nanoparticles in the solvent with time of use?   

Response:

Yes, we observed a signal from in EPR spectrum recorded for the post-reaction solution (not presented in the manuscript). It confirmed the presence of traces of TEMPO radicals in the solutions after the oxidation reaction; it looks that with every next cycle, TEMPO comes off the silver, but this process is slow, and only a tiny part is lost during every cycle. Since the reason of this process is the low oxidation of sulphur in the Ag-S bond,  probably another way of connecting TEMPO with silver would be helpful to give higher stability N-AgNPs. However, so far Ag-S bond is recognized as the best in stabilising silver and gold nanoparticles.

While the yields and TON decreased with cycle number, the TEM pictures show that the nanoparticles grew by a factor of about 20x  (from about 5 nm to more than 100 nm)... and so surface area and availability of active sites may be the reason for lower yields and TON with cycle number.  However, is the TEMPO contained within the larger particles or did it cleave off of the nanoparticles and is contained in the solution? 

Response:

Yes, that is true.

The size and shape of nanoparticles change under the reaction conditions. However, TEMPO still is connected with their surface (we confirmed this using EPR spectroscopy), so they are still active in catalysis, and TON decreases with cycle number because the grafting density of TEMPO decreases. But our catalyst system allows the selective oxidation of alcohol for several cycles and reduces the required amount of catalyst in this way.

- Can the authors explain better the reason or mechanism for nanoparticle growth in acetonitrile at (presumably) room temperature?   Did TEMPO or oxygen catalyze the sintering of the nanoparticles?  Is there anything that can be modified to prevent or reduce the rate of nanoparticle sintering? 

Response:

We think that the main reason of nanoparticles' growth is not enough stability of Ag-S bonds in the reaction system leading to slow loss of ligand molecules from the surface. A combination of TEMPO with nanoparticles via a polymer linker could likely prevent or reduce the rate of nanoparticle sintering.

Regarding the conditions of the oxidation reactions:

At what temperature was the reaction? I assume it was conducted at room temperature.

Response:

Reactions were performed at room temperature, and such information was included in the footnotes under Tables 2-4 (abbreviation rt).However, indeed it should be given in the procedure description, so we added information in the Materials and Methods section as follows:

The reactions were performed at room temperature.

"What are the roles of the other constituents in the reaction?" 

Response:

Besides the organocatalyst, (bpy)CuI complex and N-methylimidazole are used in the Stahl oxidation. According to the studies performed by Stahl and co-workers [1] in the presence of oxygen, the complex of (bpy) CuII with TEMPO and NMI is an oxidizing agent for alcohol. The formed in this process (bpy) CuI complex with NMI regenerates TEMPO from TEMPO-H. So, generally (bpy)CuI/II and NMI participate in organocatalyst regeneration.

 In our manuscript, we cite this paper describing the mechanism studies as a reference [18].

[1] 1.   Hoover, J. M.; Ryland, B. L.; Stahl, S. S. J. Am. Chem. Soc. 2013, 135, (6), 2357-2367.

"Was the reaction rate limited by oxygen diffusion into the solvent? According to Franco and Olmsted (1990), the solubility of oxygen in acetonitrile is 2.42 mM.  Will O2 interact with the other constituents in the reaction mixture?" 

Response:

Undobdetely, the solubility of oxygen in the reaction solvent is here crucial. Thus, acetonitrile was chosen because the solubility of reagents and oxygen is high in acetonitrile. Since the reaction solution is stirring very fast in an open vial, the reaction rates were rather not limited by oxygen diffusion into the solvent. We didn't perform experiments confirming such conclusions. It should be analysed using, for example, an oxygen electrode. Thank you for this suggestion; it is inspiring for planning further investigations.

"Is this oxidation process and catalyst recovery process scalable?

Considering the reactions' conditions, we do not see any serious problems with scaling up the process of alcohol oxidation. The reactions are performed under atmospheric pressure, at room temperature and using air as an abundant, cheap and non-toxic oxidising agent source. The developed synthesis of N-AgNPs allows this product to be obtained on a gramme scale; in our opinion, the oxidation reactions can also be scaled up.

As stated by the authors:  

"As shown, AgNPs can also catalyse the oxidation of alcohols by oxygen adsorption on silver surfaces [43]. Thus, we expected that using AgNPs grafted with TEMPO as an oxidation catalyst should result in a synergistic effect between these two components."

Did the authors compare the oxidation of the alcohols using the Ag-TEMPO catalysts with just the Ag nanoparticles and with just TEMPO?  What control experiments were conducted to assess the roles of Ag nanoparticles vs TEMPO on the partial oxidation of alcohols? Was any synergism observed?"

Thank you so much for this comment.

Yes, we performed control experiments with free TEMPO with the same concentration as in the case when it is grafted onto nanoparticles to analyse whether synergism exists. The results of these experiments are presented in Supporting Information (Table S3). At first sight, no synergism occurs (comparable yields were noticed for N-AgNPs and free TEMPO). However, considering that the immobilized TEMPO should exhibit lower effectiveness than free TEMPO (diffusion rate is significantly slower) and that we do not observe a decrease in yield after immobilization of TEMPO, we can conclude that the synergistic effect between AgNPs and TEMPO occurs.

We decided to introduce our conclusions on synergism in the revised version of the manuscript as follows:

Notably, the obtained yields are the same as in the employment of free TEMPO radical with the same amount as immobilised onto nanoparticles (see Table 3 and Table S3 in the Supplementary Information). Thus, it shows that the designed way of nitroxide immobilisation did not deteriorate its activity in the catalytic system. It may also indicate the synergistic effect between AgNPs and TEMPO.

Reviewer 3 Report

In this article, the authors synthesized a densely grafted nitroxide radicals to achieve highly selective oxidation of primary alcohols. The easily implementable one-pot and one-phase reaction guaranteed the economic way to get the organic catalyst. And the highly-yield, recyclable selective oxidation of different primary alcohols gave a promising prospect in future industrial production. And the mild oxidation of alcohols toward aldehydes with oxygen in the air made the whole process environmentally friendly and sustainable, which lead to a green catalyst. Here are some questions I have:

1.     The statistic diameter from DLS is much larger than the one in TEM. You mentioned it was the stabilization layer with the solvation layer, why the layers occupied such larger space than the NPs?

2.     You mentioned a significant yield decrease during the second cycle with pure oxygen or air balloon. The excess oxygen might lead to the oxidase of AgNPs when you used pure oxygen, and the ligand might be lost. Do you have any methods to prove your hypothesis? Plus, when you used air balloon the yield still dropped dramatically, why the yield could still recover to 75 % even higher than the yield in air atmosphere in the 5th cycle if the ligand lost was the only reason?

3.     What are the large shadows in your TEM when you did the catalysis after 11th cycle?

Author Response

Response to Reviewer #3's comments:

Thank you very much for your thorough reading of our work. We are very grateful for all Your comments and suggestions. Our manuscript is significantly better thanks to the changes/corrections made according to Your comments and suggestions.

  1. The statistic diameter from DLS is much larger than the one in TEM. You mentioned it was the stabilisation layer with the solvation layer, why the layers occupied such larger space than the NPs?

Response:

Thank you for the comment.

Indeed, the difference between the average diameter of metal cores (5.3 ± 1.6 nm) and particles from DLS (22.1 ± 3.96 nm) is significant.  However, the protecting organic layer in good solvent (acetone) is built of ligand molecules not constrained, also interacting with solvent molecules. In our opinion, these phenomena can explain why this layer is so thick.  

  1. You mentioned a significant yield decrease during the second cycle with pure oxygen or air balloon. The excess oxygen might lead to the oxidase of AgNPs when you used pure oxygen, and the ligand might be lost. Do you have any methods to prove your hypothesis? Plus, when you used air balloon the yield still dropped dramatically, why the yield could still recover to 75 % even higher than the yield in air atmosphere in the 5th cycle if the ligand lost was the only reason?

Response:

Thank you for the comment.

The stability of nanoparticles is a big challenge since they tend to aggregate due to high surface energy. The best solution in the case of gold and silver nanoparticles (so far) is protecting their surface by stabilising ligand molecules connected via thiolate bonds (relatively high stability). Nevertheless, Ag-S and Au-S bonds are sensitive to a high oxygen concentration. The destabilization of protecting layer leads to ligand loss.

The following text and references have been introduced in the revised version of the manuscript.

A continuous and significant decrease in yield in the presence of pure oxygen indicates that an excess of oxygen irreversibly oxidises N-AgNPs. These facts can be explained by the oxidation of sulphur in the nanocatalyst leading to the break of the Ag-S bond between nitroxide ligand and silver surface. Therefore in the consecutive cycles density of nitroxides grafted onto the nanoparticles decreases, and consequently, the yield of the selective oxidation of alcohol also decreases. In the case of other sources of oxygen in which its content is significantly lower, the sulphur oxidation rate is lower, and the decrease in yield is also considerably lower. It was reported that thiol-capped nanoparticles are stable in air for a long time, but under an oxygen atmosphere, ligands are released due to sulphur oxidation [49].

[49] Ling, X.; Schaeffer, N.; Roland, S.; Pileni, M.-P. Langmuir 2015, 31, (47), 12873-12882.

Additionally, oxidation of silver surface under the influence of high oxygen concentration may also be significant in an aspect of nanoparticles' stability.

Yes, we confirmed the presence of traces of ligand in the post-reaction solution using EPR spectroscopy (not presented in the manuscript).

Regarding the yields in the air atmosphere, ligand loss is very low in the first cycles; as we wrote, a significant meaning has a procedure of nanoparticles recovery as they adsorb very strong to surfaces, especially plastic surfaces. 

  1. What are the large shadows in your TEM when you did the catalysis after 11th cycle

Response:

This shadow visible in the TEM image is organic matter. Most probably, organic compounds are present in the reaction system ((bpy) Cu, NMI). After the 11th cycle, the nanoparticles separated from the post-reaction solution are covered with organic matter more as a ligand (TEMPO) density is lower, the surface is more available, and they are adsorbed very strongly.

Round 2

Reviewer 1 Report

The authors have corrected / implemented the manuscript following the suggestions of the reviewers, it can now be accepted for publication.

Reviewer 2 Report

The manuscript revisions provided by the authors are sufficient.